# Can Sound Alone Act as a Virtual Barrier for Horses? A Preliminary Study

**DOI:** 10.3390/ani12223151

**Published:** 2022-11-15

**Authors:** Wiktoria Janicka, Izabela Wilk, Tomasz Próchniak, Iwona Janczarek

**Affiliations:** 1Department of Horse Breeding and Use, Faculty of Animal Sciences and Bioeconomy, University of Life Sciences in Lublin, 20-950 Lublin, Poland; 2Institute of Biological Basis of Animal Production, Faculty of Animal Sciences and Bioeconomy, University of Life Sciences in Lublin, 20-950 Lublin, Poland

**Keywords:** horse, sound, virtual fencing, grazing management, motivation, perception

## Abstract

**Simple Summary:**

The concept of virtual fencing involves containing animals in a restricted area without a physical barrier by using audio and electric stimuli. Electrical pulses may initiate public concerns, and therefore, alternative aversive stimuli should be tested. As prey animals, horses are particularly perceptive to environmental stimuli. We assumed that a self-applied acute alarming sound may act as an invisible barrier for horses. Thirty horses moved individually through a designated corridor towards food or social reward when the sound was played at one of three distances (30, 15 or 5 m) from a defined line. In the case of food reward, the virtual barrier had an 80% success rate, making the horses respond with flight, going away or stopping. However, the possibility of social interactions reduced the efficacy to 20%. The sound exposure distance did not influence the barrier effectiveness but varied cardiac response with the highest stress level for a distance of 5 m. In general, sound alone has a limited potential as a deterrent and therefore cannot be applied independently. The motivation to move and the sound exposure distance are important factors for sound barrier effectiveness and its impact on the welfare of the horses.

**Abstract:**

Virtual fencing is an innovative alternative to conventional fences. Different systems have been studied, including electric-impulse-free systems. We tested the potential of self-applied acoustic stimulus in deterring the horses from further movement. Thirty warmblood horses were individually introduced to a designated corridor leading toward a food reward (variant F) or a familiar horse (variant S). As the subject reached a distance of 30, 15 or 5 m from a finish line, an acute alarming sound was played. Generally, a sudden and unknown sound was perceived by horses as a threat causing an increase in vigilance and sympathetic activation. Horses’ behaviour and barrier effectiveness (80% for F vs. 20% for S) depended on motivator (F/S), while the cardiac response indicating some level of stress was similar. The motivation for social interactions was too strong to stop the horses from crossing a designated boundary. Conversely, the sound exposure distance did not vary the barrier effectiveness, but it differentiated HRV responses, with the strongest sympathetic activation noted at a distance of 5 m. Thus, the moment of a sound playback has important welfare implications. Due to the limited potential of sound as a virtual barrier, auditory cues cannot be used as an alternative for conventional fencing.

## 1. Introduction

Virtual fences have been developed as an alternative to conventional fencing in order to comprise the trade-off between optimal grazing, costs, time and labour investments [1]. Since the system provides an enclosure without a physical barrier, it is characterized by fencing flexibility [2]. It has the potential to protect environmentally sensitive areas and brings benefits to wildlife conservation [3]. Over the last four decades, multiple types of fenceless systems have been studied, and several of them have been patented [4]. The targeted deterrent to prevent the animals from crossing a defined line can vary (e.g., audio, vibration, light) [5], but virtual fencing usually involves electric shock [6]. The most common system uses collar-mounted GPS and works by pairing an audio tone with an electric stimulus. As the animal approaches the previously established boundary, the collar emits an audio cue. Further forward movement results in receiving an electrical pulse, however, stopping or turning around prevents an aversive stimulus [7]. As shown by Marini et al. [8], the sheep’s responses were directed precisely at the auditory cue since they quickly reused the previously restricted area once it was included again. Campbell et al. [4,9] made similar observations for cattle. This is an important aspect in terms of an efficient grazing management tool [8]. Nevertheless, it cannot be excluded that animals memorise the locations where they received an aversive stimulus [2].

Although virtual fencing is not 100% effective in manipulating animal behaviour [10], many studies indicate its high potential in keeping animals in a designated area [1,2,4,7,8]. In the initial period after the introduction of an invisible barrier, the animals cannot avoid receiving the electrical pulse, and acute stress symptoms may occur. In order to meet welfare requirements, it is necessary that the animals quickly learn how to respond to the virtual fence [3]. However, individual differences were found in the rate of adaptation to the system as manifested by the different number of shocks the animal receives [11]. Although many studies show the level of stress during the learning period is comparable to that associated with many everyday procedures, the application of virtual fencing may still raise public concern for potential welfare issues [3]. Additionally, using electric shock collars for dogs is banned in some countries, and this law may also be extended to other species [12]. For these reasons, alternative repellent methods are also studied. In this case, sounds of different origins may serve as aversive stimuli [6,13,14,15].

Sound may act as an acute alarming or irritating stimulus, resulting in various behavioural reactions—rapid or delayed, respectively [6]. The volume of the sound may be stable [14] or changeable depending on the distance from the defined line. The closer the animal is to the boundary, the louder the sound is [15]. The audio cue may be emitted from a device attached to the animal [14,15] or from loudspeakers placed in a designated line [6]. The use of sounds as independent aversive stimuli has certain limitations related to the risk of habituation [13] and lower efficiency compared to traditional virtual barriers [6,14]. Nevertheless, Umstatter et al. [6] showed that auditory cues may reduce the time spent in excluded zones.

Virtual fences are mainly designed and tested for cattle [3] and, to a lesser extent, small ruminants [1,8]. Even if some of the patents targeted horses [16], there is a lack of scientific studies in this field. Ungulates can threaten biodiversity through sprig gnawing, rapid seedlings eating and bark stripping. Moreover, comfort behaviour such as rubbing against trees may result in the destruction of vegetation [17]. But it is not clear if sound alone may act as a repellent for grazing horses. Horses evolved as prey animals and have therefore developed a range of adaptations to facilitate predator avoidance [18]. Even though under domestic conditions the risk of predation is relaxed [19], horses experience many stressful situations on a daily basis. Different procedures, novel sounds or objects may all trigger the motivation to flee [20]. High sensory sensitivity is vital for controlling the surroundings and reacting quickly when necessary [21]. In the wild, horses use visual, auditory and olfactory cues to detect a predator [22]. With regard to the localisation of potential danger, the ability to hear high frequencies is particularly important [23]. Although predators are generally silent while hunting, many species respond to their vocalisation, as it directly indicates the presence of a predator [22,24]. Studies on predator recognition in domestic ungulates show they still exhibit antipredator behaviour when exposed to predators’ odours, sounds or visual signs [25,26,27] and thus, these cues may have some potential to modify the grazing activity of prey [28]. Aflitto and DeGomez [29] suggest that acoustic stimuli may act as repellents if they are biologically relevant. However, according to the risk-disturbance hypothesis [30], animal reactions towards different frightening stimuli may derive from a generalised antipredator defence mechanism. In the previous study [31], we observed that horses responded mostly to the sounds of unusual animals, like pigs or chimpanzees, and not to the vocalisation of predators. Rochais and Hausberger [32] also noted that novelty and unexpectedness provide the distraction potential of the sounds. In prey species, responses to suddenness are probably even more pronounced than those triggered by novelty, given the similarity to moving predators [33]. This may be indicative of the repellent potential of acute alarming sounds in grazing management.

Due to the lack of direct knowledge in the field of developing virtual fences for horses, we conducted a preliminary study to determine the potential of a self-applied acoustic stimulus to deter a horse from further movement. Referring to the antipredatory responses of the horses [20], an acute alarming sound effect was chosen [12]. Because horses remain vigilant to environmental stimuli and employ fast threat detection [33], we hypothesised that sound alone may act as an invisible barrier. However, we predicted that the behavioural and cardiac responses of the horses would depend on the resources available beyond the designated boundary (food vs. social reward) and on the moment of the sound.

## 2. Materials and Methods

### 2.1. Animals and Housing

The study involved 30 adult warmblood horses aged 6–16 years (15 mares and 15 geldings). All of the horses were kept in individual box stalls in the same horse riding centre for at least two years. Each box was bedded with wheat straw and equipped with a crib, hay feeder, salt lick and an automatic waterer. The subjects were fed a mixture of oats and bran three times a day and had unlimited access to hay. They were used as leisure horses and worked under the saddle for a maximum of 2 h a day in the afternoon or evening hours. The horses were released to an expansive pasture or several paddocks for at least six hours a day. They were all familiar with the pasture and the pre–pasture paddock (experimental paddock) leading towards it. The subjects were under permanent veterinary control and were constantly observed by an experienced caretaker during daily handling. No physical or behavioural disorders were found before or after the study.

### 2.2. Procedures of the Sound Tunnel Test

#### 2.2.1. General Conditions and Test Preparation

The virtual barrier potential of the auditory stimulus was assessed during the sound tunnel test (STT) that was conducted in spring 2022. It consisted of playing back a recording of a futuristic sound while a horse moved towards a known pasture (providing locomotory, grazing and social opportunities) through a designated corridor (sound tunnel; ST). The ST was set up five days before the start of the study to let the horses get familiar with the novelty in their environment for two hours a day. Prior to testing, the horses were familiarised with the movement direction through the ST, with five runs being completed. The pasture was separated from the experimental paddock (pre-pasture paddock), where the ST was placed, by a permanent fence of metal railings. However, the tested horse was able to remain in visual and auditory contact with the herd staying at an average distance of approximately 200–400 m from the tunnel. While one of the horses was subjected to STT, none of the other horses involved in the study was in the pasture at the time to avoid the risk of habituation to the experimental sound. They all stayed in the stable building located about 300 m from the experimental paddock while waiting for the test. After completing the STT, the horse was brought to the paddock situated next to the stable. After the last horse had completed the test, all individuals were led to the pasture. The experiment was not carried out on windy (<0.3 m/s) or rainy days to reduce the risk of interfering with the perception of the sound stimulus.

The horses were brought to the experimental paddock in the morning hours by a known caretaker. After 5 min of the rest, they were individually introduced to the ST—a 55 m long and 4 m wide corridor designated with 1.55 m plastic poles and tape for an electric fence, commonly used to demarcate a grazing area (Figure 1a,b). It was open-closed from one side (start) and opened on the end side (finish), letting a horse roam freely after it finished a test. The first 5 m was the ‘start zone’, where the subject waited for the test to begin while being held by a known caretaker. Yellow cones were placed every 10 m outside the tunnel to ease the observer’s control of a horse’s movement through the corridor from a distance of 10 m. Additionally, three green cones were used to mark a start and finish line and the place where the sound was played as the horse reached it (sound exposure distance).

#### 2.2.2. Carrying Out the Test

The horses were randomly divided into three experimental groups of 10 subjects depending on the sound exposure distance and comprising one of the following distances from the loudspeaker placed at the finish line:First group—30 mSecond group—15 mThird group—5 m

Each group included five mares and five geldings (to minimise the risk of the gender impact on the results) and was exposed to the sound at the same distance throughout the study. The sound–indicating cone (the green cone) (Figure 1b) was relocated over one of three distances to facilitate the experimenter playing the sound on time. Each horse was subjected to two tests on two separate days: Variant F—the motivator (additionally to a near pasture) for movement was a food rewardVariant S—the motivator (additionally to a near pasture) for movement was a familiar and friendly horse that was not included in the study (social reward)

Prior to the study, we interviewed the horse keeper. The caretaker indicated a horse that was the most friendly towards the other individuals and the least inclined to agonistic behaviour. The interval between the two variants was one week. The food reward was placed 3 m behind the end of the ST, whereas a known conspecific stood around 70 m further to minimise the sound impact on it. Additionally, it had been habituated to the experimental sound before the study. The familiar horse was not attached or confined to allow the horses to have social contact after each test. However, to prevent locomotory behaviour during tests, the familiar horse was provided with attractive food while the first horse was being tested. 

Fifteen randomly selected horses were subjected to the STT in the following order: variant F and then variant S. The other individuals were tested in reverse order to reduce the risk of the order of variants affecting the horses’ reactions (Table 1). Five horses were tested on a given day.

Each variant (F, S) of the STT consisted of three stages: A—recalling the direction of movement and showing the reward; no measurementsB—a control trial; behavioural measurementsC—an experimental trial; behavioural and cardiac activity measurements

During stage A, a horse was led by a known caretaker from the start to the finish line of the ST, where it received a reward (variant F: food, variant S: social contact). If the individual went actively and willingly with a caretaker in the second trial, stage A was considered completed. The horses needed 2–4 repetitions of the first stage. In stage B, the caretaker released the subject from the rope in the start zone. The horse was al-lowed to move through the tunnel on its own. It was then brought back to the start zone again. In stage C, which immediately followed stage B, a sound was introduced while the horse moved through the tunnel on its own. Horses were freely released for the next 5 min after the sound was played.

#### 2.2.3. Sound Stimulus

The futuristic sound used in the study was played with a wireless speaker (JBL Charge 4, rated power of 30 W) connected to a Samsung Galaxy A02s device (Samsung Electronics Co., Ltd., Suwon-si, Republic of Korea) via Bluetooth. It let the experimenter play the sound from a distance of 10 m from the ST, where he stayed during the test. The 20-s recording was prepared in Audacity 2.4.2. software by combining short fragments of several seconds into one and amplifying the entire recording by 12 dB. The sound was violent and novel for subjects and was played at a sound intensity level (LA) of approximately 80 dB measured at a distance of 1 m from the loudspeaker to provide its acute alarming features [12,13]. This intensity has been proven to have only a short-term effect on the cardiac response [31]. If the horse reached the finish line before the end of the recording time, the recording was stopped.

### 2.3. Behavioural Data Collection and Analyses

The time of STT completion and the time or frequency of certain behaviours were recorded separately for stages B and C for both variants F and S of the study (Table 2). Two dependent variables, i.e., ‘barrier effect’ (the first reaction to the sound scored on a 4-point scale indicating a sound barrier effect) and ‘latency time’ (latency time to respond to the sound), were determined only for stage C. Furthermore, ‘latency time’ was measured only for the horses that stopped, went away or ran away in response to the sound stimulus. Because the time needed to complete an STT varied between the horses, behaviours measured per unit time [s] (locomotory behaviour, vigilance) were calculated as a ratio of a certain behaviour to the tunnel completion time [%] and were used in this form in the statistical analysis. If the horse did not leave the tunnel within 5 min, the test was considered finished, and the percentages of individual behaviours were calculated in relation to 300 s.

### 2.4. Cardiac Activity Data Collection and Analysis

The emotional arousal of the horses in response to the sound barrier was measured based on the parameters of heart rate variability (HRV) using Polar Vantage M telemetric devices. An elastic strap with electrodes (Polar Pro) and a transmitter (H10) attached to it was fastened around the horse’s chest so that the electrodes were adjacent to the left side of the horse at the heart level. To optimize electrode-skin contact, the inner side of the electrode part of the strap was covered with ECG gel. Horses were familiarised with the equipment for three subsequent days before the start of the study. On each experimental day, devices were put on the horses in the stable at least 10 min before the test. The devices were synchronized with electronic timers to enable controlling the moment of sound playback and then to monitor heart activity changes during the successive periods of the test. Recordings were carried out continuously and stopped 5 min after the sound was played. Due to the character of the study, cardiac responses were collected during the moderate movement. To facilitate the assessment of the sound impact on the emotional arousal of the horses, heart rate recordings were analysed in three 1—minute subsequent periods [36]: directly before the sound (starting parameters—pre-sound period; S1), from the moment of the sound playback (sound period; S2) and during the fifth minute after the sound was played (post-sound period; S3). The 1-min HRV measurements were chosen due to the short-lasting character of the test and the sound stimulus used (20 s) and were proved to provide a reliable assessment of the parameters of HRV [36]. A 3-point measurement allowed to monitor the immediate (S2) and delayed (S3) effects of the sound on the cardiac responses of the horses. Heart function was measured with HR (bpm) and heart rate variability (HRV) parameters that reflect the modulating effect of the autonomic nervous system on cardiac activity [37,38]: -RMSSD (ms)—standard deviation of differences between successive IBIs (time domain),-HF (ms^2^)—high-frequency component (0.07–0.5 Hz) of HRV determined by spectral analysis (frequency domain),-LF (ms^2^)—low-frequency component (0.005–0.07 Hz) of HRV determined by spectral analysis (frequency domain),-LF/HF (%)—low frequencies/high-frequencies ratio (frequency domain).

An increase in HR reflects sympathetic activity, in LF and LF/HF reflect a mix of sympathetic and vagal influences towards a sympathetic shift, whereas RMSSD and HF increases directly reflect parasympathetic (vagal) activity. Laborde et al. [36] strongly recommend drawing conclusions based on HRV indices that reflect clearly identified physiological systems with a theoretical underpinning, such as parameters of the vagal tone being measured through RMSSD (in the time domain) and HF (in the frequency-domain). Thus, RMSSD and HF (to check whether the results are consistent) were taken as the main variables. LF and LF/HF were used to additionally discern differences between the impact of the studied factors on the reactions of the horses. Heart rate recordings were analysed with Kubios HRV Standard 3.5.0 software. Very low threshold filters of beat correction were used to eliminate artefacts when necessary.

### 2.5. Statistical Analyses

The statistical analysis was performed with SAS 9.4 software [version 9.4 by SAS Institute Inc., Cary, NC, USA]. The behavioural data, but not HRV data, showed deviations from normality assumptions (checked by Kolmogorov-Smirnov and Shapiro-Wilk tests). For this reason, non-parametric tests were used for behavioural variables, whereas parametric tests were conducted for HRV measurements. Non-parametric analyses were performed using a Wilcoxon assessment, a Kruskal-Wallis test and Dwass, Steel and Critchlow-Fligner multiple comparisons. Statistical analyses for HRV were conducted using the procedures of the general linear model and the least mean squares. In the case of behavioural analysis, the qualitative variables included the combinations of traits: test variant (F or S reward)_stage of the study (B or C) and distance from the speaker (30, 15 or 5 m)_stage of the study (B or C). For better clarity, the variables were presented and discussed as mean values ± standard deviation (SD). Only comparisons within the same variant (FB–FC, SB–SC) or distance (5B–5C, 15B–15C, 30B–30C) or between different variants (FB–SB, FC–SC) or distance (5B–15B–30B, 5C–15C–30C) within the same stage were pictured and discussed as other comparisons (e.g., FB–SC, 5B–15C) were not the subject of the interest. In the case of cardiac measurements, the independent variables were: the variant of the study (F or S), the distance from the loudspeaker (30, 15 or 5 m) and the stage of the sound tunnel test (B or C). The interactions of ‘the variant’ or ‘the distance’ and the stage of the study were analysed. In addition to the main factors, the differences between mares and geldings were checked during the STT. For all analyses, the level of significance was set at *p* < 0.05. Only significant results are presented in the results section.

## 3. Results

Prior to the experiment, the heart rate variability resting parameters that were to be assessed during the study were measured. Mares were characterized by higher values of the HR (39.0 bmp) and LF (2170.5 ms^2^) compared to geldings (37.7 bmp and 1797.1 ms^2^, respectively). Except for HR (higher values for mares), there were no differences in behavioural and cardiac variables between mares and geldings during the STT (*p* > 0.05). For this reason, the factor ‘gender’ was excluded from further analysis.

### 3.1. Behavioural Responses

The variant of the study indicating the motivation to move through the ST (food or social reward) had a greater impact on behavioural responses of the horses to the unknown sound than the distance at which the horse was exposed to it (30, 15 or 5 m from the loudspeaker) (Table 3). Regarding the studied variant (F or S), significant differences were observed in four compared to one behavioural variables for the effect of the sound exposure distance.

#### 3.1.1. The Variant of the Study

Except for vigilance (higher during variant F; *p* < 0.05), horses reacted similarly (*p* > 0.05) during the control trial (B) both when the reward was food and social contact (Table 3). Depending on the studied variant, significant changes in behaviour between stages B and C were noted (*p* < 0.05). Horses needed more time to complete an STT (C: 99.67 ± 97.95 s, B: 31.97 ± 20.27 s), spent less time on total locomotion (C: 68.57 ± 25.07%, B: 94.57 ± 6.28%), were more vigilant (C: 18.60 ± 14.29%, B: 2.29 ± 3.98%) and snored more frequently (C: 4.03 ± 4.42, B: 0.60 ± 1.99) when the motivation to move forward was food. When individuals moved towards a familiar horse, only the percentage of time spent alert (C: 5.02 ± 12.32%, B: 0.00 ± 0.00%) changed significantly. The time required to get through the tunnel and the time of remaining vigilant were relevantly longer during variant F compared to variant S of the study (*p* < 0.05). Additionally, horses spent less time on locomotion and snored more when provided with food than social reward (*p* < 0.05).

#### 3.1.2. Sound Exposure Distance

Regarding the distance to the finish line at which a horse was exposed to the experimental sound (Table 3), only vigilance increased significantly in stage C within each distance (5 m: 17 times, 15 m: 10 times; 30 m: 8 times; *p* < 0.05). However, there were no differences in the duration of standing alert between three distances during stages C and B of the study (*p* > 0.05).

#### 3.1.3. The Sound Barrier Effect

Overall, in the two variants of the study, the sound barrier was 50% effective (30 out of 60 desired reactions). The virtual barrier effect (stopping, going away or running away after the sound appearance) was noted in the case of 24 horses in variant F (success: 80%) of the STT, whereas only six animals responded with stopping or going away towards the start zone during variant S (success: 20%) (Table 4). Regarding success rate, horses gained more points in behavioural scale (the sound barrier effect was more pronounced) when they had no possibility of direct social contact (F: 3,21 ± 0,98 vs. S: 2,33 ± 0,52; *p* < 0.05) after the STT completion (Figure 2). The time of this response did not differ between the test variants (*p* > 0.05). An inverse relationship was observed for the effect of the sound exposure distance. The latency time was relevantly longer (*p* < 0.05) at a distance of 30 m from the loudspeaker in comparison to 15 and 5 m (about 3 and 5 times, respectively), but no differences in the barrier effect of the sound were noted (*p* > 0.05). However, there was a trend that horses tested at 30 m reacted more calmly (Table 4).

### 3.2. Cardiac Activity Responses

Although all included cardiac indices changed during the sound tunnel test, shifts towards vagal or sympathetic activation differed depending on the HRV parameters (Figure 3 and Figure 4). When changes were observed, LF and LF/HF, as well as RMSSD and HF, dropped gradually, both when regarding the motivator (Figure 3) and the sound exposure distance effect (Figure 4). It was also noted that heart rate changes (HR) were immediate but short-term (significant growth in S2 period and then drop in S3 period), whereas changes in HRV parameters were delayed (drop in S3 compared to S1; gradual drop or S2 period; rapid drop)—they did not differ between sound and pre-sound period in any case.

#### 3.2.1. The Variant of the Study

There were no differences (*p* > 0.05) in pre-sound (S1), sound (S2) or post-sound (S3) measurements between the two test variants (Figure 3). Only changes within the variants were observed (*p* < 0.05). 

HR increased rapidly just after the sound appearance and dropped relevantly in the fifth minute after sound exposure in comparison to S2 and S1 in both variants of the study. RMSSD decreased between S3 and S2 (a delayed, but rapid drop), while HF decreased between S3 and S1 (delayed, gradual drop) during variant F of the study. In the case of variant S, both RMSSD and HF values were characterized by a delayed, gradual drop. There was a deferred decrease in LF value in S3 compared to the S1 period when the horses were provided with a food reward, but no changes in LF/HF were noted. On the contrary, LF and LF/HF dropped rapidly between S3 and S2 measurements when the horses were allowed direct social contact. Among HRV indices, LF (variant S) and LF/HF (both variants) increased after the sound was played, but these changes were not statistically significant (*p* > 0.05).

#### 3.2.2. Sound Exposure Distance

Considering the effect of the sound exposure distance (Figure 4), the values of the HRV parameters during the pre-sound period did not differ significantly in most of the cases (*p* > 0.05). The only differences were found between 5 and 30 m for LF and between 5 and 15 m for LF/HF, with higher values for 30 m and 15 m, respectively. Similarly, only single differences were noted in corresponding measurements between different distances during the sound period (*p* < 0.05). The highest increase in HR was noted for the distance of 5 m. The lowest values of RMSSD after sound playback were found for the horses tested at a distance of 5 m, but they were significantly lower than the values for the 15 m distance exclusively. In turn, LF during the S2 period measured at the distance of 15 m was significantly higher than those for 5 m and 30 m. In the post-sound period, the only relevant difference concerned RMSSD, which was meaningfully lower (*p* < 0.05) for the horses exposed to the sound at 5 m (72.34 ± 43.35 ms) compared to 15 m (109.08 ± 64.81 ms).

Within the variant of the study, the direction of HR changes was similar to the food and social reward variants of the study (Figure 3). A rapid increase in HR during the sound period and then a significant drop (*p* < 0.05) in the post-sound period were observed for each distance (Figure 4). LF and LF/HF did not change during the test in the case of horses from 5 m group (*p* > 0.05). A drop (*p* < 0.05) in both parameters was noted for 30 m (LF: S3 vs. S1, LF/HF: S3 vs. S2), and a drop in LF was also visible for 15 m (S3 vs. S2). RMSSD and HF changed significantly only in the horses tested at a 5 m distance. In both cases, a delayed drop of these parameters in S3 compared to the S1 period (*p* < 0.05) was noted. In the case of RMSSD (15 m), LF (15 m) and LF/HF (5 and 30 m), the increase was observed during the S2 period, but these changes were not statistically significant (*p* > 0.05).

## 4. Discussion

The current study revealed that a non-biologically relevant acute alarming sound is generally perceived by horses as threatening and, therefore, triggers avoidance or flight responses in the opposite direction. Similar observations were made by Umstatter et al. [14], who found that cows exposed to audio stimuli turned away from a designated line. These results show that self-applied acoustic stimulus may have some potential to manipulate the locomotion and dispersion of horses on pastures. However, a general success rate of 50%, which is in agreement with the results obtained by Umstatter [14], eliminates the commercial use of the sounds as independent virtual fences. Umstatter et al. [6] came to the same conclusions in the previous study by testing irritating sounds to control the location of cows. For these reasons, acoustic stimuli may be rather used as a support for conventional fencing. Similar recommendations were made for a traditional (paired with electric shock) virtual fencing system to reduce the risk of animals accessing roads or public areas [3]. We observed that the efficiency of the sound barrier decreased from 80% when the motivator to cross the boundary was a food reward to 20% in the case of social reward. This clearly indicates that a sound alone, even frightening, is too weak to stop a horse from further movement when it is highly motivated to obtain certain resources. Another concern is the risk of rapid habituation to sounds, which was observed by Butler et al. [13] and which would further reduce the potential of audio stimuli as independent barriers. In the current study, we did not investigate this factor due to its limited implications (weak potential of sound barrier). However, habituation should be an important factor when testing the effectiveness of virtual fencing and attempts to reduce this effect should be investigated. Due to the high vigilance and perceptual sensitivity of horses [21,33], even subtle changes in sound playback may have a significant impact on reducing the rate of habituation. 

The time needed to complete a sound tunnel increased significantly only when the food reward was provided, whereas delaying further walking is crucial for the barrier effect. Moreover, Jouven et al. [10] noted that the effectiveness of the virtual fence for sheep declined in the presence of conspecifics, despite the previous association of the audio cue with an electrical pulse. It is commonly known that horses are highly motivated to any kind of social interaction [39,40], which was also observed in our study. Considering only successful trials, there were no differences in latency time to stop and go or run away in response to the sound in both variants of the study. If horses reacted to the sound, their response was just as quick. This fact has an evolutionary origin, as fast threat (predator) detection ensured survival [41]. Regarding the prolonged time of sound tunnel completion, the percentage of time spent on locomotory behaviour decreased after the sound application in the food reward variant of the study. This provides some potential for auditory stimuli as a grazing management tool. Using the sound barrier made the horses more vigilant, which is thought to be a primary antipredator defence mechanism [28], but the time of standing alert was much shorter in the presence of the second horse. Moreover, the frequency of snoring, acting as stress-releasing behaviour to restore the homeostasis [42], increased only during the food reward variant of the study. Altogether, these results may suggest that social motivation acts simultaneously as a social buffer that helps horses face a frightening stimulus [43]. Furthermore, the tested subjects observed a calm and habituated companion, which may reduce fear reactions in stressful situations significantly more than a non-experienced companion [44]. This finding is important in a social context but limits the possibility of using sounds as independent invisible barriers. However, the current study does not provide an answer to how horses exposed to a sound barrier in a group would react. Social impact on horses’ readiness to cross the sound boundary was visible in behaviour, but no differences in HRV parameters were found between the two variants of the study. The trend in changes in cardiac activity in both cases was similar. 

In general, regarding both main study factors (motivator and sound exposure distance), the moment of the sound appearance had a significant influence only on HR responses. HRV indices were mostly characterized by a gradual drop in their values. HRV reflects the modulating effect of ANS on cardiac responses [37]. In the previous study [31], we showed that a 1-min recording was not sufficient to influence HRV responses (RMSSD, LF, HF, LF/HF), and mainly only HR and RR changes were noted. In that particular case, horses were in the known paddock in a group and were not supposed to border a line. In the current study, horses were encouraged to move forward when a sudden, frightening sound appeared. These circumstances had an impact on HRV changes, but the response was delayed. Another reason for these differences may be the measurement time—5 min in the previous [31] and 1 min in the current study. 

Considering the food and social reward variants of the study, the RMSSD and HF decrease in the post-sound period indicates a decrease in vagal tone, hence, higher sympathetic activity a few minutes after the sound appearance [37]. Simultaneously, lower values of LF (both variants) and LF/HF (social reward variant) in the post-sound period reflect some relaxation in SNS. This variation in HRV parameters may be due to the ‘conflicting’ character of the test. Although an aversive stimulus induced a stress response, obtaining the reward mitigated its negative effects. As shown by Safryghin et al. [45], the physiological response depends on the context that evoked the changes. An HR increase was seen both in positive and negative contexts. Thus, some conflicting contexts may have been reflected in HRV changes in our study. In terms of the companion of a second horse, a slight increase in LF/HF (activation towards SNS) was observed directly after sound application, followed by a rapid decrease in the fifth minute after auditory stimulus appearance. This result may suggest the importance of the presence of a companion in alleviating stress symptoms [44]. 

Besides emotional arousal, an HR increase in the sound period might also have been influenced by a flight reaction, which was noted in 14 out of 60 cases. However, even if it occurred, escape in trot or canter was usually limited to a few seconds. Since gaits faster than walking occur in nature as a defence against predators [46], a horse’s response to the acute alarming sound might have been analogous. Lenoir et al. [47] observed that HR increased, while RMSSD, LF and HF decreased, during intensive, compared to moderate, physical effort. Although a gradual drop in HRV parameters was noted, the increasing intensity of movement was not a rule in the our study, which was also confirmed by the significant drop in HR values in the post-sound measurement. Thus, it suggests that changes in HRV may have been a result of a virtual barrier impact. All things considered, the impact of the changes in movement in response to a sound based on HRV parameters cannot be excluded, and this is a weakness of the current paper.

Regarding the influence of the sound exposure distance, we observed the opposite behavioural-physiological response than for the impact of a motivator. Behavioural variables did not differ between horses tested at three distances (30 m, 15 m or 5 m), whereas there was a clear tendency in HRV changes between the groups. This finding is in accordance with an experiment conducted by Safryghin et al. [45], who showed that a behavioural response may not reflect the physiological state of the organism. We observed that the only behavioural variable that depended on the sound playback was vigilance. No matter the distance, horses were more alert during the sound application test stage. Only the time to respond to the sound varied between three distances. Horses tested at 30 m reacted the slowest to the sound barrier, but still, a latency time of 2.61 ± 1.07 s was sufficient to stop the horses from further walking. Moreover, the immediate reactions of the horses to the auditory stimulus showed some tendency to be the calmest, which may have safety implications [20,33]. 

In terms of cardiac responses, the most disturbing was the sound played at a distance of 5 m. In the wild, sudden stimuli are particularly important in inducing an avoidance response [41], so audio cues played at a distance of 5 m from the horse may have been perceived as more threatening. Moreover, sounds at this distance were heard as louder, which, according to Butler et al. [13], might have resulted in stronger responses. Only HR changes towards SNS activation were visible immediately after audio cue exposure in each experimental group. The delayed decrease of RMSSD and HF values, suggesting an increasing level of stress after sound appearance, was noted only for the lowest horse–loudspeaker distance. Additionally, PNS activation was visible for greater sound exposure distances by the drop of LF (for 15 m) and LF/HF (for 15 m and 30 m) and not for 5 m, which indicated some relaxation after obtaining a reward. These results show that the possibility of detecting a frightening stimulus had a positive effect on the emotional arousal of the horses. Vigilance plays a crucial role in detecting and localising a stressor [28], and horses tested at a distance of 30 m had the most time to react. When considering the use of sounds in the developing of virtual fences or supporting traditional fencing methods, an appropriate distance triggering the sound playback should be taken into account. Regarding the efficiency of sound as a barrier and its welfare implications, we recommend longer distances in establishing an animal–loudspeaker ‘triggering point’ when using stationary speakers.

## 5. Conclusions

An acute alarming sound affected the antipredator defence mechanism of the horses, suggesting some potential as a virtual barrier. However, the motivation for social interactions was too strong for a sound alone to stop the horses from crossing a designated boundary. Due to its insufficient efficacy, the self-applied acoustic stimulus cannot be used independently. Nevertheless, its possible useas a support for conventional fencing cannot be completely excluded, e.g., in areas with a higher risk of animal escapes by increasing vigilance and redirecting of movement, In this case, further research would be needed, as the solution would require the design of a suitable and cost-effective automated system.

The sound stimulus affected the cardiac responses of the horses, indicating at least partial activation of the sympathetic nervous system. To achieve a compromise between efficiency, safety and welfare concerns, the moment of sound playback should allow the horses to detect it early enough.

## Figures and Tables

**Figure 1 animals-12-03151-f001:**
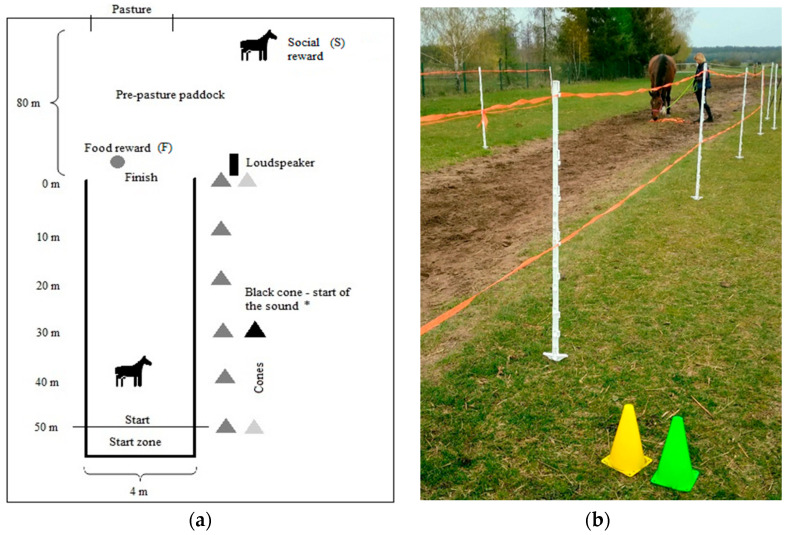
(**a**). The outline of a sound tunnel and an experimental paddock. (F), (S)—the first and second variants of the study (two separate tests); cones—subsequent 10 m distances from the loudspeaker (two cones at the start and finish); black cone—positioned 30, 15 or 5 m from the finish line (*), indicating the moment of sound playback; start zone—waiting for the test to begin. (**b**). The end part of the sound tunnel and a horse obtaining a food reward after familiarity with the tunnel.

**Figure 2 animals-12-03151-f002:**
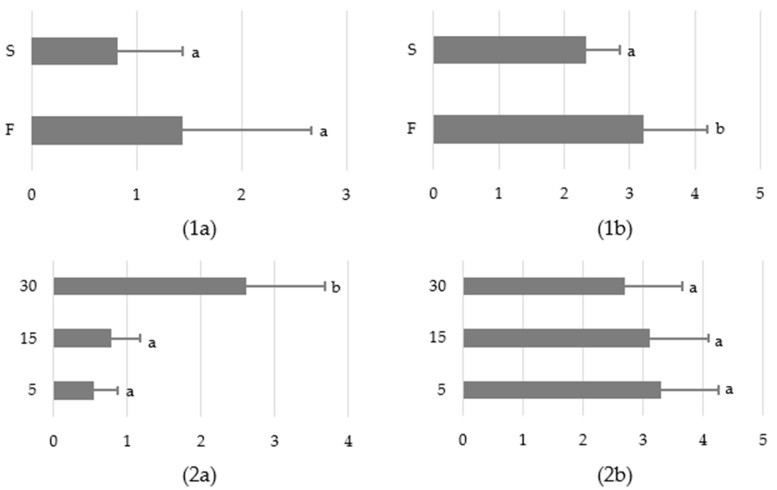
(**1a**). The impact of the test variant (food reward—F/social reward—S) on latency time to respond to the sound [s]. (**1b**). The impact of the test variant on the effect of the sound barrier assessed on a 4-point scale. (**2a**). The impact of the sound exposure distance (30, 15 or 5 m) on latency time to respond to the sound [s]. (**2b**). The impact of the sound exposure distance on the effect of the sound barrier assessed on a 4-point scale. Means marked with different letters differ significantly at *p* < 0.05.

**Figure 3 animals-12-03151-f003:**
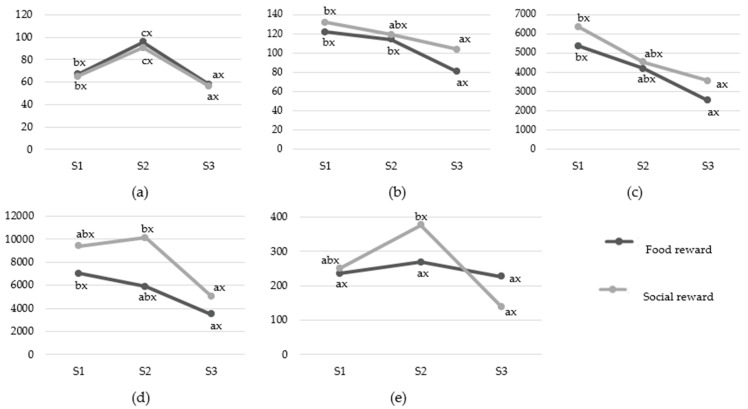
Mean values of: (**a**) HR [bpm]; (**b**) RMSSD [ms]; (**c**) HF [ms^2^]; (**d**) LF [ms^2^]; (**e**) LF/HF [%] in subsequent test periods: pre-sound (S1), sound (S2) and post-sound (S3) period depending on the test variant (motivator: food/social reward). Means marked with different letters differ significantly at *p* < 0.05 within the same variant of the study in subsequent test periods S1, S2 and S3 (letters a, b) or between different variants within the same period (letters x, y).

**Figure 4 animals-12-03151-f004:**
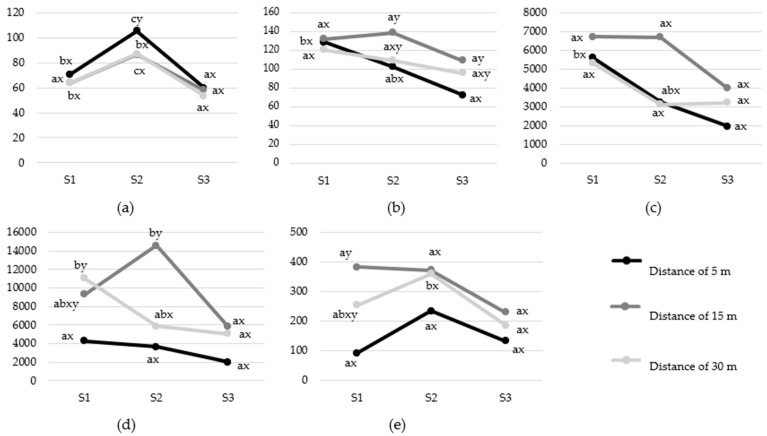
Mean values of: (**a**) HR [bpm]; (**b**) RMSSD [ms]; (**c**) HF [ms^2^]; (**d**) LF [ms^2^]; (**e**) LF/HF [%] in subsequent test periods: pre-sound (S1), sound (S2) and post-sound (S3) period depending on the sound exposure distance (30/15/5 m). Means marked with different letters differ significantly at *p* < 0.05 within the same distance in subsequent test periods S1, S2 and S3 (letters a, b) or between different distances within the same period (letters x, y).

**Table 1 animals-12-03151-t001:** The order of the test variants.

Day	Variant	Group of the Horses	Day	Variant	Group of the Horses
Week 1			Week 2		
1	Variant F	a	7	Variant S	a
2	Variant S	b	8	Variant F	b
3	Variant F	c	9	Variant S	c
4	Variant S	d	10	Variant F	d
5	Variant F	e	11	Variant S	e
6	Variant S	f	12	Variant F	f

**Table 2 animals-12-03151-t002:** Behaviours assessed during stages B and C of each test variant (F, S).

Behaviour	Description
Tunnel completion [s]	Time taken by the horse to cross the tunnel from the start line to the finish line
Walk/trot/canter [%]	Time spent walking/trotting/cantering
Locomotion [%]	Total time of locomotion (walk, trot and canter)
Vigilance [%]	Time spent standing alert; standing still with elevated neck, intently orientated head and pinnae [34]
Snoring [freq.]	Total number of a very short raspy inhalation sound [35]
* Defecation [freq.]	Total number of repetitions of the elimination of faeces
** Barrier effect [scale 1–4]	The first reaction to the sound; 1—continuing forward movement, 2—stopping, 3—going away, 4—running away
** Latency time [s]	Latency time to respond to the sound; assessed only for horses that stopped, went away or ran away in response to the sound stimulus

%—percentage of time spent on a certain behaviour regarding the total time of tunnel completion; freq.—the number of repetitions, * behaviour eliminated from the analysis due to the variability of results close to or equal to zero, ** variables assessed only in stage C.

**Table 3 animals-12-03151-t003:** Differences in horses’ behaviour (means ± SD) in two stages of the sound tunnel test (B—control trial, without the sound and C—experimental trial, with the sound), depending on the test variant and sound exposure distance.

Behavioural Variable	Study Factors	Stage of the Study	
		B	C
Test variant (food—F/social—S reward)			
Tunnel completion time [s]	F	31.97 ± 20.27 ax	99.67 ± 97.95 by
	S	25.27 ± 15.90 ax	27.07 ± 25.45 ax
Total locomotion [%]	F	94.57 ± 6.28 bx	68.57 ± 25.07 ax
	S	98.50 ± 5.01 ax	93.50 ± 13.21 ay
Vigilance [%]	F	2.29 ± 3.98 ay	18.60 ± 14.29 by
	S	0.00 ± 0.00 ax	5.02 ± 12.32 bx
Snoring [freq.]	F	0.60 ± 1.99 ax	4.03 ± 4.42 by
	S	0.33 ± 1.30 ax	0.60 ± 2.30 ax
Sound exposure distance (5/15/30 m from the speaker)			
Vigilance [%]	5	0.48 ± 2.13 ax	8.29 ± 10.38 bx
	15	1.28 ± 3.58 ax	13.69 ± 18.75 bx
	30	1.67 ± 3.20 ax	13.45 ± 18.75 bx

Means marked with different letters differ significantly at *p* < 0.05 within the rows (letters a, b) or within the columns for the same behavioural variable (letters x, y).

**Table 4 animals-12-03151-t004:** Number of the horses that ignored, stopped, went or ran away after the sound application regarding the variant of the study and the sound exposure distance.

Study Factors	Ignoring	Stopping	Going Away	Running Away
Variant of the study				
Food reward	6	9	1	14
Social reward	24	4	2	0
Sound exposure distance				
5 m	10	3	1	6
15 m	10	4	1	5
30 m	10	6	1	3

## Data Availability

The data presented in this study are available on request from the corresponding author.

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
