# Peer review of "Can Sound Alone Act as a Virtual Barrier for Horses? A Preliminary Study"

_animals, 2022, doi:10.3390/ani12223151_

Round 1
Reviewer 1 Report
Introduction
This study is well justified. As the authors highlight the use of virtual fencing is becoming more common in various livestock industries. The use of electric shocks as part of the electric fencing is not appropriate in horses, and therefore it is worthwhile investigating whether audio stimuli alone can limit horse movement. The introduction addresses the important issues that are relevant to the study.
Materials and methods
The study design, data collection methods and statistical analysis were appropriate. The description of the study design is a little difficult to follow in places and could be made clearer by appropriate editing.
Results
The results are a little difficult to understand because of the way in which they are presented. The captions for the figures are also a little unclear, and it may be that the data presented would be more easy to understand as a table.
Conclusion
The content of the conclusion is appropriate based on the results presented and the rationale for the study outlined in the introduction. I question the author’s conclusion that audio barriers could be used as a support for conventional fencing. What would be the advantage of providing audio support? And of the experimental conditions the audio stimulus was triggered by one of the researchers but presumably for general use an automated system would be required that would, in all likelihood, be expensive to implement.
Overall
The authors set out to answer a specific question and I believe that they have done that successfully with an appropriately designed study. Based on the experimental results they describe there appears to be little value in the use of an audio stimulus is a virtual fence for horses. There is no reason to support some sort of role for an audio barrier if none exists. The manuscript is difficult to read in places, probably due to the fact that English is likely to be the authors’ second language-if that is the case they are to be congratulated for the quality of their presentation. However that does not mean that the manuscript would not be improved by appropriate editing.
Author Response
ANSWERS TO REVIEWER 1
REVIEVER: Materials and methods
The description of the study design is a little difficult to follow in places and could be made clearer by appropriate editing.
AUTHOR:
Thank you for your comment. We tried to improve the description of the study design by dividing subsection 2.2. Procedures of the sound tunnel test into three clearly separate parts that, we hope, will ease a reader to follow it. Additionally, we distinguished the different research groups and factors by bullet points. We also changed the order of some sentences or edited them.
REVIEWER: Results
The results are a little difficult to understand because of the way in which they are presented. The captions for the figures are also a little unclear, and it may be that the data presented would be more easy to understand as a table.
AUTHOR:
Indeed, some figures may be difficult to analyze them. We decided to present a part of behavioural data (where two factors were shown and compared with a, b and x, y letters (fig. 2, fig. 3)) as a table. However, we did not change the way of presentation the data showing sound barrier effect (fig. 4). We believe that this figure is transparent, as each of its parts (1a, 1b, 2a, 2b) presents comparison for only one factor (only letters a, b), but we tried to improve the caption. Even though figures showing HRV changes (fig. 5, fig. 6) concern two factors (letters a, b and x, y), we think its better to show them in this form, not as tables. Line graph that we used to present these data shows the process of HRV changes in subsequent measurements, what, in our opinion, ease readers to perceive and understand these differences. Compared to behavioural data in fig. 2 and 3, its easier to compare differences between subsequent periods (S1-S3), as letters a and b are on the same graph line, and between the same period for different groups (e.g. S1 between food and social reward), as letters x and y are always one below the other. However, we tried to improve figures’ captions to make them more clear. In fact, some of them might have been confusing. According to these changes, we corrected the entire text reffering to a changed numbers of tables and figures.
REVEWER: Conclusion
I question the author’s conclusion that audio barriers could be used as a support for conventional fencing. What would be the advantage of providing audio support? And of the experimental conditions the audio stimulus was triggered by one of the researchers but presumably for general use an automated system would be required that would, in all likelihood, be expensive to implement.
AUTHOR:
Thank you for your precious remark. The way that we introduced this reflection may be a little bit misleading and it definitely must be explained. By support for conventional (but often not effective) fencing, we meant sound barrier effect on horses’ behaviour (increase in vigilance, stopping or redirection of movement) and consequently, its possible potential to reduce for example a risk of escaping in some problematic places, e.g. like roads. We named this effect as “support” to emphasise that when used for the purpose described above, it can only be used in combination with a physical fence. In this case, certain automated system would be needed, but this was only a preliminary study to assess the potential of sounds as virtual barrier. We decided to emphasise that this potential would require further research. According to the Revewer’s comment, we improved conclusions section and abstract.
REVIEWER: Overall
The authors set out to answer a specific question and I believe that they have done that successfully with an appropriately designed study. Based on the experimental results they describe there appears to be little value in the use of an audio stimulus is a virtual fence for horses. There is no reason to support some sort of role for an audio barrier if none exists. The manuscript is difficult to read in places, probably due to the fact that English is likely to be the authors’ second language-if that is the case they are to be congratulated for the quality of their presentation. However that does not mean that the manuscript would not be improved by appropriate editing.
AUTHOR:
Thank you for your precious comments that much helped to improve our paper. When it comes to English, before the submission our article was checked and revised by a native speaker, for which we can provide a certificate.

Reviewer 2 Report
The topic of this study is interesting and unusual, in many aspects. Authors evaluated the efficacy of virtual fences as alternative to conventional ones, to deter herds of horses from moving in specific areas. Many aspects of this innovative strategy have been taken into consideration, not least a low impact on the skyline and naturalistic environments (preserving plants and trees from defacement of rubbing, munching or stomping). Looking at the other side of the coin, virtual fences often use, specific stimulus (acoustic or electric) designed to induce a flight response as a deterrent for animals and, as a consequence, this stimulus may trigger a stress reaction. To control for this possibility, authors monitored Heart Rate Variability(HRV). It must be said that HRV’ indices have been carefully considered here, paying attention to the recent and updated literature for their interpretation.
However no relevant data emerged: even though, results are well discussed by showing a deep knowledge of horses’ evolutionary specialization, the study just tell the readers that an acoustic fence doesn’t work for horses. The manuscript results as a test to proof a theory, not really adding substantial news on horses’ reactions to stimuli.
Just few concerns:
- Social reward: how did the authors measured the level of affiliation between horses?
- Line 348 the sentence “horses more violently (gained more points in the behavioral scale)” may be softened (I would not use “violently”) and clarify (since the only feature characterized by a scale of the behaviors considered is the barrier effect, I suggest writing it)
- Line 182-183 authors stated that the familiar horse used for “social reward” was not confined to allow horses to have social contact after each test. The potential frustration accumulated during test would be in this way tempered by contact with a conspecific. Later in line 349 and also in line 393 authors stated that horses were allowed or not (depending on the line) to have indirect social contact. How should “indirect social contact” be defined in this case?
- Line 461: reference numb. 42 corresponds to a review. In this particular statement however, it should be useful to cite the study demonstrating how snores are used to release stress, which is Scopa, C., Palagi, E., Sighieri, C., Baragli, P., 2018. Physiological outcomes of calming behaviors support the resilience hypothesis in horses. Sci. Rep. 8, 17501; also present in the review
- In my opinion the great absentee in conceptualizing this study is not testing for habituation. Of course, since horses did not restraint to access specific area although the disturbing sound, testing habituation would have been pointless and impracticable. However, in the big picture, one should hypothesize even how to prevent habituation.
Author Response
ANSWERS TO REVIEWER 2
REVIEWER:
The topic of this study is interesting and unusual, in many aspects. Authors evaluated the efficacy of virtual fences as alternative to conventional ones, to deter herds of horses from moving in specific areas. Many aspects of this innovative strategy have been taken into consideration, not least a low impact on the skyline and naturalistic environments (preserving plants and trees from defacement of rubbing, munching or stomping). Looking at the other side of the coin, virtual fences often use, specific stimulus (acoustic or electric) designed to induce a flight response as a deterrent for animals and, as a consequence, this stimulus may trigger a stress reaction. To control for this possibility, authors monitored Heart Rate Variability(HRV). It must be said that HRV’ indices have been carefully considered here, paying attention to the recent and updated literature for their interpretation.
However no relevant data emerged: even though, results are well discussed by showing a deep knowledge of horses’ evolutionary specialization, the study just tell the readers that an acoustic fence doesn’t work for horses. The manuscript results as a test to proof a theory, not really adding substantial news on horses’ reactions to stimuli.
AUTHOR:
Thank you for you precious observations and comments.
REVIEWER:
Social reward: how did the authors measured the level of affiliation between horses?
AUTHOR:
All of the tested horses were familiar with each other and were kept in the same horse riding centre for at least two years, which is described in subsection 2.1. Animals and housing. They were familiar with a pasture where they all were released during day hours. Prior to the experiment, we additionally interviewed the caretaker. He indicated a horse that was the most friendly to each other and the least likely to engage in agonistic behaviour. Thank you for your remark, we now completed this information in our paper (description of test variants in subsection 2.2. Procedures of the sound tunnel test).
REVIEVER:
Line 348 the sentence “horses responded more violently (gained more points in the behavioral scale)” may be softened (I would not use “violently”) and clarify (since the only feature characterized by a scale of the behaviors considered is the barrier effect, I suggest writing it)
AUTHOR:
Thank you for your advice. We introduced the appropriate changes to the text, referring to the Reviewer’s comments.
REVIEWER:
Line 182-183 authors stated that the familiar horse used for “social reward” was not confined to allow horses to have social contact after each test. The potential frustration accumulated during test would be in this way tempered by contact with a conspecific. Later in line 349 and also in line 393 authors stated that horses were allowed or not (depending on the line) to have indirect social contact. How should “indirect social contact” be defined in this case?
AUTHOR:
Thank you for your perceptiveness and observation. Honestly, it is a spelling mistake. Both in line 349 and 393 a word “indirect” is supposed to be replaced with “direct”, as horses were allowed to make a physical contact. In turn, indirect contact (like visual contact with a herd on a pasture) was allowed throughout the study in all cases. These mistakes were now corrected in the text.
REVEWER:
Line 461: reference numb. 42 corresponds to a review. In this particular statement however, it should be useful to cite the study demonstrating how snores are used to release stress, which is Scopa, C., Palagi, E., Sighieri, C., Baragli, P., 2018. Physiological outcomes of calming behaviors support the resilience hypothesis in horses. Sci. Rep. 8, 17501; also present in the review
AUTHOR:
Thank you for your advice, we improved this sentence and changed the reference.
REVIEWER:
In my opinion the great absentee in conceptualizing this study is not testing for habituation. Of course, since horses did not restraint to access specific area although the disturbing sound, testing habituation would have been pointless and impracticable. However, in the big picture, one should hypothesize even how to prevent habituation.
AUTHOR:
This is a valuable observation. We agree that it would be pointless and impracticable to test habituation in the case of our study, where we showed a low effectiveness of sound barrier. However, we also agree that we should extend our paper to consider the problem of habituation. We raised this issue in discussion section, showing the weak sign of our paper and emphasising the problem of habituation when using sound barriers.

Round 2
Reviewer 2 Report
All suggestions have been addressed.